# Human local field potentials in motor and non-motor brain areas encode upcoming movement direction
Etienne Combrisson [1,2,3] ✉, Franck Di Rienzo [2], Anne-Lise Saive [1,4], Marcela Perrone-Bertolotti [5], Juan L. P. Soto[6], Philippe Kahane [7], Jean-Philippe Lachaux[8], Aymeric Guillot [2] & Karim Jerbi [1,9,10] ✉

Limb movement direction can be inferred from local field potentials in motor cortex during movement execution. Yet, it remains unclear to what extent intended hand movements can be predicted from brain activity recorded during movement planning. Here, we set out to probe the directional-tuning of oscillatory features during motor planning and execution, using a machine learning framework on multi-site local field potentials (LFPs) in humans. We recorded intracranial EEG data from implanted epilepsy patients as they performed a four-direction delayed center-out motor task. Fronto-parietal LFP low-frequency power predicted hand-movement direction during planning while execution was largely mediated by higher frequency power and low-frequency phase in motor areas. By contrast, Phase-Amplitude Coupling showed uniform modulations across directions. Finally, multivariate classification led to an increase in overall decoding accuracy (>80%). The novel insights revealed here extend our understanding of the role of neural oscillations in encoding motor plans.

The direction of arm movements can be inferred from the firing pattern of individual neurons in the primary motor cortex (M1)[1,2]. Critically, the firing rate of M1 neurons has been shown to depend on the movement direction, a well-established phenomenon known as directional tuning. A directionally tuned neuron exhibits maximum firing rates during arm movement in its "preferred direction" and gradually lower rates for other directions. Most studies have used single unit activity (SUA) to decode movement parameters in non-humans[3–6] or humans[7].

A growing body of research suggests that multi-unit activity (MUA) and Local Field Potential (LFP) signals can also be used to predict movement directions from the monkey motor cortex[8–12]. Further evidence in macaques has revealed clear task-related LFP modulations in the gamma band in the posterior parietal cortex during both the planning and execution of arm and eye movements[13,14]. In humans, LFP-based movement type identification and directional tuning have been investigated with intracranial EEG (iEEG) data acquired during pre-surgical evaluations in patients with drug-resistant epilepsy[11,15–19]. Interestingly, when comparing the decoding power achieved by the different frequency components of the iEEG signal, these studies show that the highest directional tuning is often found in the low-pass filtered signals (<4 Hz) and in the power of the so-called broadband gamma band (~60–140 Hz). This corroborates directional tuning findings reported in monkeys[9,16,20] and non-invasively in humans[21]. The ability to infer movement type and direction from invasive and non-invasive recordings has a direct clinical application for brain–computer interfaces[22–24].

The above studies provide compelling evidence that limb movement direction can be decoded using spectral power properties of both invasive and non-invasive motor cortex recordings during movement execution. However, it is still not clear whether other frequency-domain features, such as oscillation phase and cross-frequency interactions, exhibit directional tuning. Furthermore, little is known about the temporal dynamics with which movement direction is represented in the brain during planning. To address these gaps, we investigated whether the classification of arm movement directions was possible using phase, amplitude, and phase–amplitude coupling (PAC) features during both planning and execution. We then asked

¹Psychology Department, University of Montreal, Montreal, QC, Canada. ²University of Lyon, UCBL-Lyon 1, Laboratoire Interuniversitaire de Biologie de la Motricité UR 7424, F-69622 Villeurbanne, France. ³Institut de Neurosciences de la Timone, Aix Marseille Université, UMR 7289 CNRS, 13005 Marseille, France. ⁴Cognitive Science Department, Lyfe Research and Innovation Center, Ecully, France. ⁵Université Grenoble Alpes, Université Savoie Mont Blanc, CNRS, LPNC, 38000 Grenoble, France. ⁶Telecommunications and Control Engineering Department, University of Sao Paulo, Sao Paulo, Brazil. ⁷Université Grenoble Alpes, Inserm, U1216, CHU Grenoble Alpes, Grenoble Institut Neurosciences, GIN, Grenoble, France. ⁸Lyon Neuroscience Research Center, EDUWELL team, INSERM UMRS 1028, CNRS UMR 5292, Université Claude Bernard Lyon 1, Université de Lyon, F-69000 Lyon, France. ⁹Mila (Quebec AI Institute), montreal, QC, Canada. ¹⁰UNIQUE Centre (Quebec Neuro-AI research Center), Montreal, QC, Canada. ✉e-mail: e.combrisson@gmail.com; Karim.Jerbi.UdeM@gmail.com

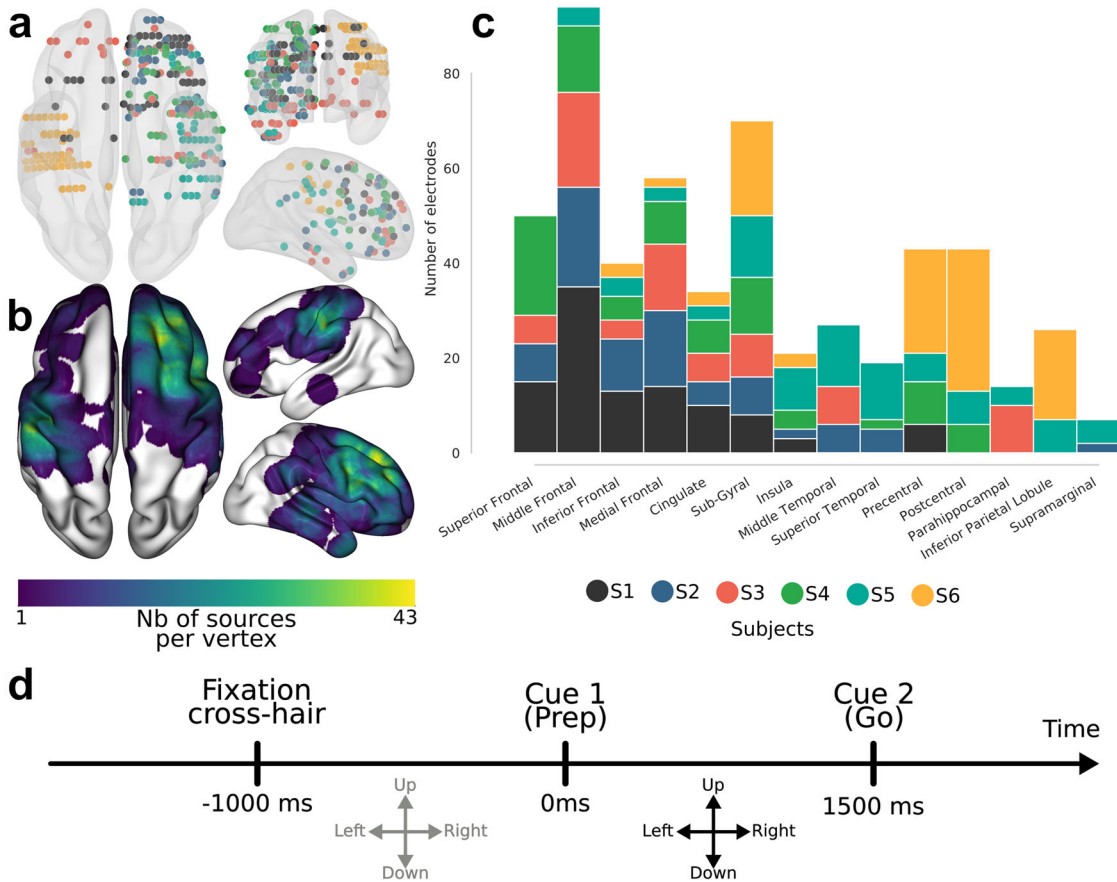

**Fig. 1 | Representation of intracranial implantation and brain coverage across subjects projected on a standard 3D MNI brain and experimental design. a** Top, front and right views of the depth-electrode recording sites. **b** Top, left and right views of the number of recording sites that contribute to each vertex (i.e. spatial density). **c** SEEG locations per subject **d** Experimental design of the delayed center-out motor task. After a 1 s rest period (*rest*, −1000 to 0 ms) a first cue (Cue 1) instructed subjects to prepare to move their hand (*motor planning*, 0–1500 ms). Next, a go signal (Cue 2) appeared prompting participants to execute the movement (*motor execution*, 1500–3000 ms).

whether movement directions share common neural representations during both phases. To tackle this question, we trained a classifier at execution and tested whether it was able to decode movement directions at planning and conversely. Investigating the cross-temporal generalizations in both directions was aimed at probing similarities of neural representations of movement directions between planning and execution. Importantly, we were able to explore this question using depth recordings from over 700 cortical sites across six epilepsy patients. Local field potentials were continuously monitored using stereotactic-EEG (SEEG) while participants performed a classical delayed center-out motor task. We hypothesized that movement decoding should be possible from the moment movement planning starts, i.e. during the delay period preceding movement onset (0–1500 ms). In addition, based on previous work emphasizing the importance of phase and PAC in motor tasks[25–28], we also hypothesized that these features, alongside spectral power, would display directional tuning.

Our results provide evidence for successful prediction (up to 86%) of intended arm movements in humans, using combinations of oscillatory phase and power features extracted from LFPs in motor and non-motor structures. Single-feature direction decoding revealed the prominent role of alpha oscillations and broadband gamma activity during planning and execution, respectively. Finally, our multi-site electrophysiological decoding framework reveals insights into the temporal dynamics of movement encoding.

## Results

In the following we present the findings of decoding arm movement directions in human epileptic patients implanted with intracranial electrodes and performing a center-out motor task (Fig. 1). We begin with (i) the

results of single-feature classification of movement directions, followed by (ii) insights on the temporal evolution and generalization of the decoding, and finally (iii) the multivariate classification results. But, first, we illustrate the features probed in this study; Figs. 2–4 illustrate the extracted features for a representative premotor electrode, and they show the temporal evolution including (i) pre-stimulus rest, (ii) planning, and (iii) execution windows for the amplitude and phase features. The PAC illustrations are shown using co-modulograms for each direction in both planning and execution periods. These represent the features that were computed for 748 sites from all participants in this study.

### Decoding movement directions: single features findings

Significant decoding of movement direction during both planning and execution periods using power and phase features were prominent in motor areas (i.e. supplementary motor area (SMA), premotor cortex (PMC) and primary motor (M1)) and parietal brain regions showed (Fig. 5). During the planning period, the highest decoding accuracy of 49.37% (chance level 25%, $p < 0.05$) was found using alpha power in the posterior middle frontal gyrus (pMFG), anterior middle temporal gyrus (aMTG), posterior cingulate and ventral precuneus. The anterior pre-SMA and ventral precuneus also showed significant decoding using alpha power during the execution period. The spatial distribution of significant decoding during planning using beta power was similar to the decoding patterns obtained using the alpha band but the maximum of decoding accuracy was slightly smaller (46.25%, $p < 0.05$). During execution, hand-movement direction classification using beta power reached a maximum of 50.75% ($p < 0.05$) in the PMC. High-gamma (60–200 Hz) power led to

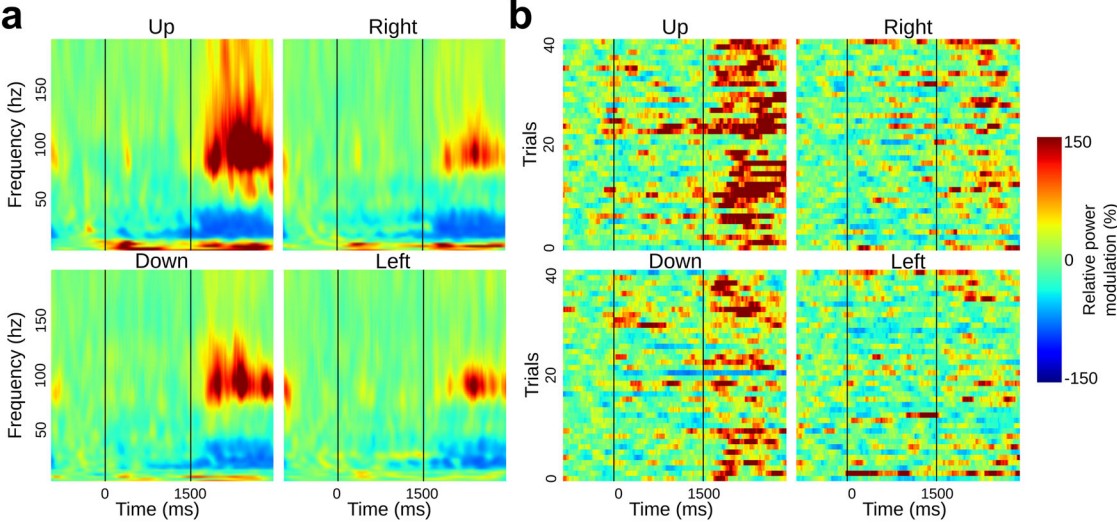

**Fig. 2 | Relative power modulations per directions (up/right/down/left) relative to baseline ([−750, −250] at rest) for a premotor SEEG site. a** Time–frequency representation. **b** Singe-trial high gamma [60, 200 Hz] power modulation.

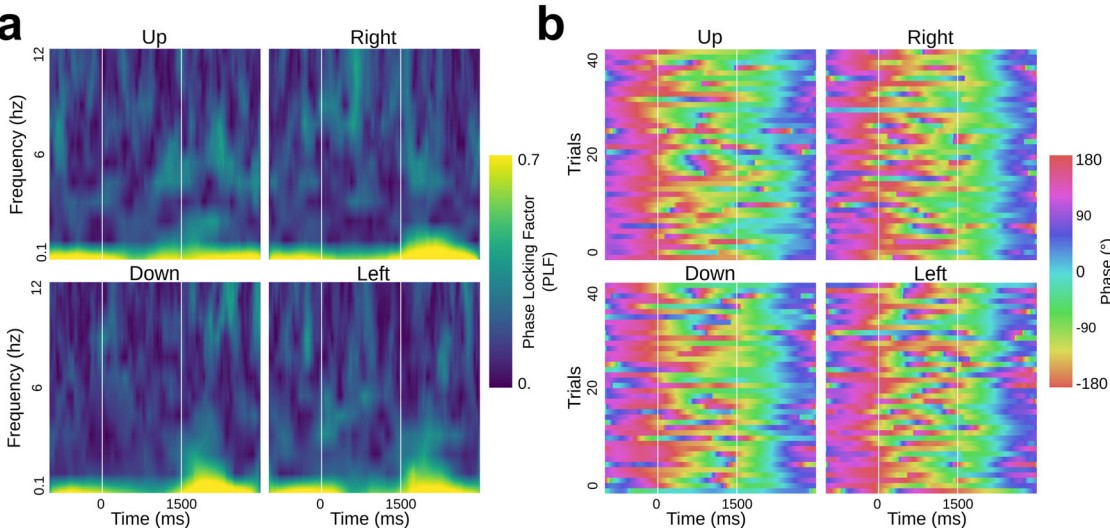

**Fig. 3 | Phase modulations per directions (up/right/down/left) for a premotor SEEG site. a** Phase-locking factor across trials. **b** Singe-trial very low-frequency phase (VLFC, [0.1, 1.5 Hz]) modulation.

62.94% ($p < 0.05$) correct classification during the execution in the posterior pre-SMA. Decoding results clearly surpassed those obtained in the lower-frequency bands. Still, during the execution, high-gamma power also revealed statistically significant decoding in M1. Interestingly, the posterior pre-SMA and the ventral precuneus also showed significant decoding for both planning and execution. Among all non-power features, only the VLFC phase exhibited brain areas with significant decoding (i.e., posterior pre-SMA) during the execution with a maximum of 44.38% ($p < 0.05$).

As far as PAC is concerned, we found an increase of alpha-gamma coupling in the dorsal sensorimotor and premotor cortices during movement planning, followed by a decrease during execution. Although subtle differences in PAC were observed across directions, the differences were not sufficient to allow for significant PAC-based movement classification.

### Decoding network associated with intended and executed limb movements

Several brain structures allowed for direction prediction at various moments in time during either planning or execution. To explore the spatial and temporal dimensions of the decoding, we computed and visualized for, at each time bin and within various regions of interest (ROIs), the number of sites with statistically significant decoding (Fig. 6). This figure was obtained using a univariate gaussian kernel estimate (kdeplot from the Python package Seaborn). To isolate the features that presented decoding robustness across time, we kept only features with significant decoding in at least three consecutive time bins after correction for multiple comparisons ($p < 0.05$). Through the task, alpha power was the first feature to enable decoding during the planning period starting in the pMFG [150, 1150 ms] and almost simultaneous in the ventral precuneus [250–450 ms]. Interestingly, this last structure is the only one that also revealed a second window of decoding during the execution [2250, 2600 ms]. Gamma power then took over, allowing for a decoding starting from the end of the planning period in the anterior pre-SMA [1050, 2650 ms] followed by the posterior pre-SMA [1550, 2650 ms]. It is interesting to note that around 2000 ms, the density of significant time bins decreased in the anterior pre-SMA while increasing in the posterior pre-SMA. In the PMC, the executed direction was successfully predicted between [1700, 2650 ms] using the gamma power and in an almost identical time interval [1800, 2600 ms] using the beta power.

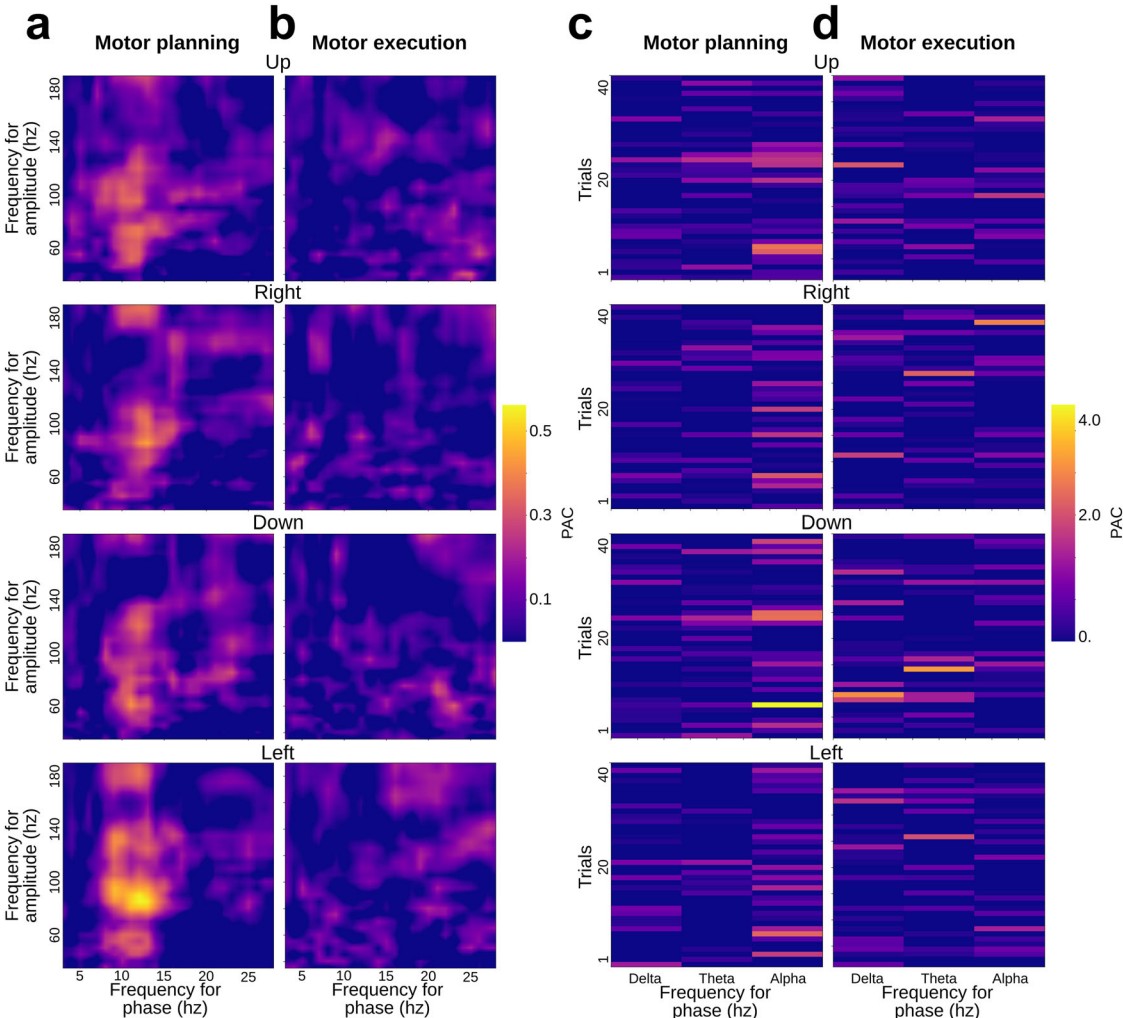

**Fig. 4 | Phase-amplitude coupling modulations per direction (up/right/down/ left) during planning [0, 1500 ms] and execution [1500, 3000 ms] windows for a premotor iEEG recording site.** Comodulograms representing PAC as a function of frequency for phase and amplitude during planning (**a**) and execution (**b**). Singe-trial PAC modulations, per direction, for delta [2, 4 Hz], theta [5, 7 Hz] and alpha [8,13 Hz] phase coupling with high-gamma [60, 200 Hz] amplitude for planning **c** and execution **d** windows.

## Time-resolved modulations and decoding

To exploit the temporal resolution of the SEEG-based decoding framework, we chose to illustrate the temporal dynamics of modulation and classification across movement direction in three key sites. Figure 7 illustrates the time-resolved directional tuning and single feature decoding in three sites, using power in alpha and gamma bands and VLFC phase. Two sites located in the pMFG (Fig. 7a, b) shared the same alpha power pattern: a uniform alpha power across the four directions during rest, followed by directional tuning during planning and finally, a similar alpha desynchronization during execution. It is worth noting that alpha power inter-trial variability (assessed by the standard error on the mean) across directions was higher during the planning, compared to the execution period. For the first site in the pMFG (Fig. 7a), the four directions were independently modulated from 300 ms after the onset of the planning period (Cue 1). The power difference across directions was maximal around 1050 ms. This was also the time associated with the maximal decoding accuracy of 43.6% ($p < 0.05$). For the second site in the pMFG (Fig. 7c), direction-specific modulations were observed earlier, around 200 ms before the onset of planning and single-feature maximum decoding of 47.4% was reached around 250 ms. Interestingly, planning horizontal movements (i.e left and right directions) were clearly dissociated with alpha power modulations going in opposite directions. Conversely, vertical movements (i.e up and down directions) seemed to be stable during planning. In comparison, the gamma power in the pre-

SMA posterior presented no direction-specific patterns during rest and planning (Fig. 7c). Instead, gamma power allowed decoding directions from 200 ms after the execution onset (Cue 2), with a maximum decoding accuracy of 62.5% at 2250 ms (i.e., 750 ms after Cue 2). Interestingly, in this same site in posterior pre-SMA, significant direction classification (46.4%, $p < 0.05$) was also achieved using VLFC phase (Fig. 7d). Similarly to gamma power, the VLFC phase happened to have a consistent behavior across directions during rest and planning phase, but showed direction-specific modulations during execution. Intriguingly, VLFC phases showed the highest differences during the execution at approximately the same time as gamma power (around 2245 ms), which led to a significant decoding.

## Temporal generalization of the decoding of movement directions

We then tested whether movement direction representation was shared between planning and execution. To this end, we used TG using either single or multiple power features (e.g., alpha alone or in combination with gamma) to find if some SEEG sites were able to decode during the execution while training the classifier during the planning period and conversely (Fig. 8). Panels a–c represent TG using single power features respectively in pMFG (alpha), and two distinct motor sites in the posterior pre-SMA (high-gamma) while panel D is the TG for those three combined sites and features. In pMFG, TG analysis showed that classifiers trained during the first 500 ms of planning were able to generalize to data from 0 to 1000 ms after Cue 1

with a maximum decoding of 48.75% using alpha power modulations (Fig. 8a). In the posterior pre-SMA, classifiers were able to generalize to data within the execution period and reached a maximum of 63.13% (Fig. 8b). This execution-related sustained neural pattern did not share enough common representation with the planning period in order to show generalization. By contrast, Fig. 8c presents another SEEG site located in posterior pre-SMA for which part of the information about hand directions was shared between planning and execution. Indeed, a classifier trained during execution provided significant decoding during execution (max. 50.63%) but also during the planning phase (max. 44.38%). Interestingly, training the classifier during the planning did not lead to significant decoding during the

execution (i.e., non-symmetrical behavior). Finally, significant decoding patterns of the three previous TGs were conserved when features were combined (Fig. 8d); Training during the execution and testing during planning reached a maximum of 55% (i.e. +4% compared to the posterior pre-SMA site only). Moreover, a distinct decoding pattern emerged when the classifier was trained during the planning phase (~200–600 ms) and tested during the execution (~2300–2500 ms).

### Decoding results of the multi-feature procedure

To explore the joint relevance of multiple features in predicting movement directions, we extended the decoding process to a multi-feature (MF) framework combining all computed feature types (power, phase and PAC), all frequency bands and SEEG sites (Fig. 9). The MF classification was carried out using feature selection (see the "Methods" section) for each of the 67 time points, leading to a distinct set of features at each time bin. The time-resolved decoding accuracy in subject S1 (Fig. 9a) yielded two clear bumps: during the movement intention phase, with a maximal decoding of 76.12% at 850 ms, followed later on by a second bump during the middle of the execution phase and a maximum decoding of 84.25% at 2050 ms. Because the decoding performance at each time bin was based on a distinct set of features, we counted the number of times each feature appeared (i.e. occurrence) during the entire planning and execution period, and we also grouped those features by Brodmann areas (BA) (Fig. 9b). The most frequently selected features were predominantly power-based but the importance of each frequency band varied between preparation and execution periods. Among all of the non-power features (i.e. phase and PAC), the VLFC phase and the coupling between delta phase and gamma amplitude (delta-gamma) were the most selected features during the execution only. For planning, slow oscillations (i.e. beta and alpha) were predominantly selected. Unlike planning, the execution was predominantly decoded using high-frequency power features. We also summarized the maximum decoding reached across subjects for both planning and execution (Fig. 9c), as well as the most frequently selected features across subjects (Fig. 9d). Both decoding of motor planning and execution reached a maximum of 86% (S1, for both hands). In general, the decoding of movements during execution was higher (or at least equal to) the accuracies obtained during planning except for subject 5. This subject presents a 14% difference (i.e. 84% for planning and 70% for execution) which probably reflects the fact that this subject has an SEEG implantation that is more suitable for intention prediction. Subjects 4 and 6 did not present significant decoding even with the multi-features procedure. The MF direction decoding during the planning phase was primarily achieved using slower frequencies (i.e. delta, theta, alpha, and beta) in premotor, prefrontal, and parietal areas (BA6-9-40). During execution, decoding directions more frequently involved high frequency power (i.e. low and high gamma), especially in pre and primary motor areas (BAs 4-6-8-11).

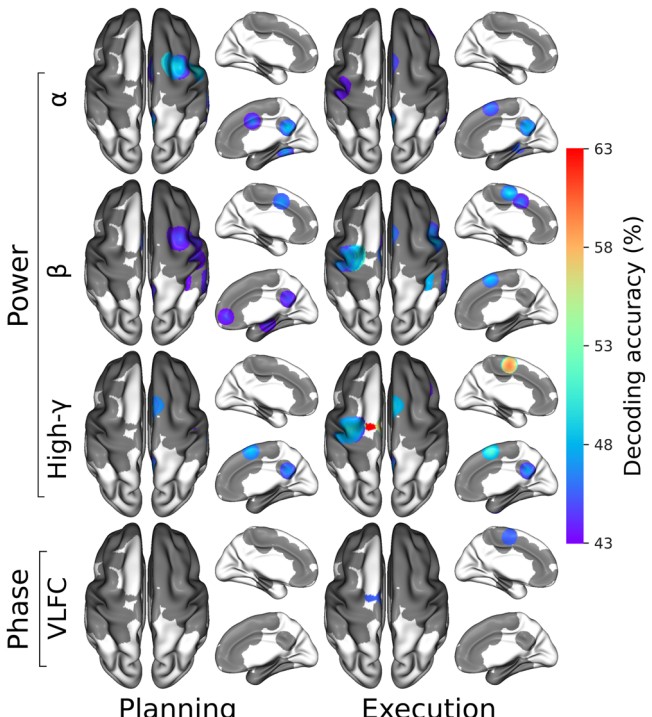

**Fig. 5 | 4-directions decoding of intended and executed limb movements using power, phase and PAC features over several frequency bands, Power features are presented within alpha (α), beta (β) and high-gamma (high-γ) bands and VLFC ([0.1, 1.5 Hz]) phase.** Each column summarized SEEG sites that present significant decodings during the entire planning or execution period. Non-significant areas are presented in gray (p < 0.05 after correction for multiple comparisons using maximum statistics through SEEG sites, time, and frequencies).

**Fig. 6 | Sorted density of significant timings across region of interest (ROI) and power features.** This density was obtained using single feature that presented at least three consecutive significant decodings after correction for multiple comparisons (p < 0.05 corrected for SEEG sites, features and time using maximum statistics).

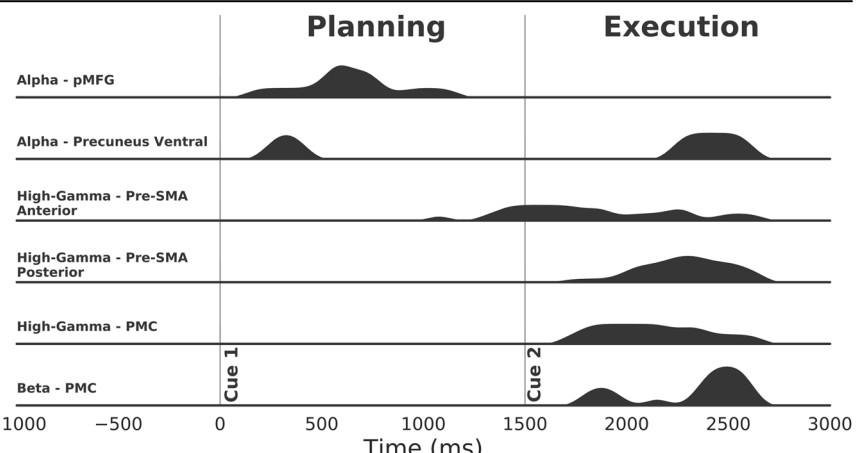

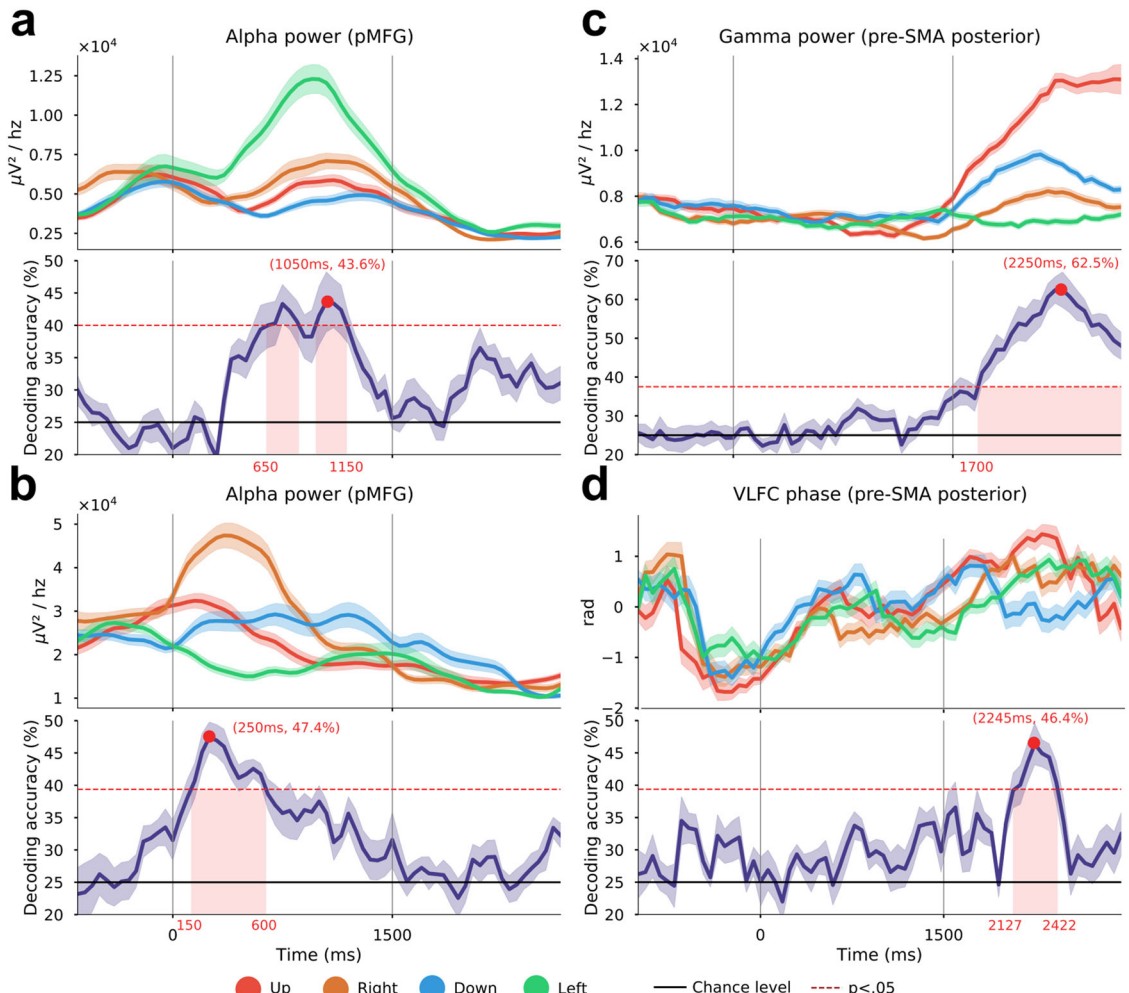

**Fig. 7 | Time-resolved 4-directional tuning task-induced power and phase modulations.** (up: red; right; brown: down: blue; left: green) and associated decoding accuracies (purple) using an LDA with a 10 times 10-fold cross-validation on three SEEG sites. The power is computed every 50 ms using a 700 ms window. The two vertical lines at 0 and 1500 ms, respectively, represent the onset of the planning phase (Cue 1) and the execution phase (Go signal, Cue 2). The horizontal black plain line represents the theoretical chance level (4-classes, 25%) and the red dotted line represents the significance level computed from permutations at $p < 0.05$ after correction for multiple comparisons through time points using maximum statistics, **a** and **b** alpha power [8, 13 Hz] for two electrode contacts located in the posterior middle frontal gyrus (pMFG), **c** high-gamma power [60, 200 Hz] of a posterior pre-SMA electrode contact, **d** VLFC phase [0.1, 1.5 Hz] of the same posterior pre-SMA site. Shaded areas represent the SEM.

## Discussion

The goal of the present study was to expand on the current understanding of how neural oscillations encode movement direction. In particular, we examined whether movements can be predicted from a range of oscillatory features recorded during or even prior to execution (i.e. in the delay period while participants waited for the go cue). To this end, we employed a machine learning approach, where the success rate of each oscillatory brain feature (or combination thereof) in predicting movement direction was taken as a measure of functional relevance to direction encoding. Alongside well-established spectral power features, the supervised learning framework we used also probed the predictive strength of phase and phase-amplitude coupling, both of which have received very little attention in the context of movement decoding. A further added value of the present study was the use of SEEG, providing multi-site LFP depth recordings in humans; This allowed us to probe distributed network decoding patterns across time and over widely distributed brain areas, not limited to primary motor regions.

Our findings show that the direction of upcoming movements can be predicted using spectral features extracted during movement planning from widely distributed human LFPs. In fact, the accuracy by which movements were predicted from neural signals acquired during the delay period (up to 86%) was equivalent to the rate of successful decoding achieved with data acquired during actual movement execution. However, the anatomical locations and main frequency bands of the LFP features that led to the best classifications during execution differed from those that allowed for the highest predictions during planning. During execution, the best features were high gamma power in motor and premotor areas, while the classification of movement intentions was mainly achieved through alpha and beta power in premotor, prefrontal, and parietal regions. From a decoding perspective, the highest decoding accuracies were obtained in a multivariate decoding framework combining multiple oscillatory signal features (e.g., power, phase, and phase–amplitude coupling) across multiple intracranial recording sites (e.g., motor, premotor, but also non-motor-areas).

The fact that oscillatory power in multiple frequency bands, especially in motor areas, carries directional information is well-established[9,21]. Additionally, previous research suggests that phase signals can be used to infer hand position, velocity, and acceleration through low-frequency components[26,29,30]. Yet, to the best of our knowledge, this is the first study to jointly explore the relative importance of amplitude, phase and PAC features for decoding planned or executed limb movement directions in humans. Our results confirm the prominent role of power features in movement direction decoding and show that very low frequency (<1.5 Hz) phase features also led to statistically significant direction prediction,

**Article**

**Fig. 8 | Temporal generalization (TG) using power features on three distinct SEEG sites.** The vertical and horizontal lines at 0 and 1500 ms stand respectively for Cue 1 and Cue 2. White contoured zones delimit statistically significant decodings at $p < 0.01$ (binomial test) after Bonferroni correction through time. No decoding are performed on the diagonal, **a** TG in a pMFG site using alpha [8, 13 Hz] power, **b** and **c** TG in two distinct SEEG sites located in the posterior pre-SMA using high-gamma [60, 200 Hz] power, **d** TG of the three combined sites (alpha pMFG + high-gamma posterior pre-SMA).

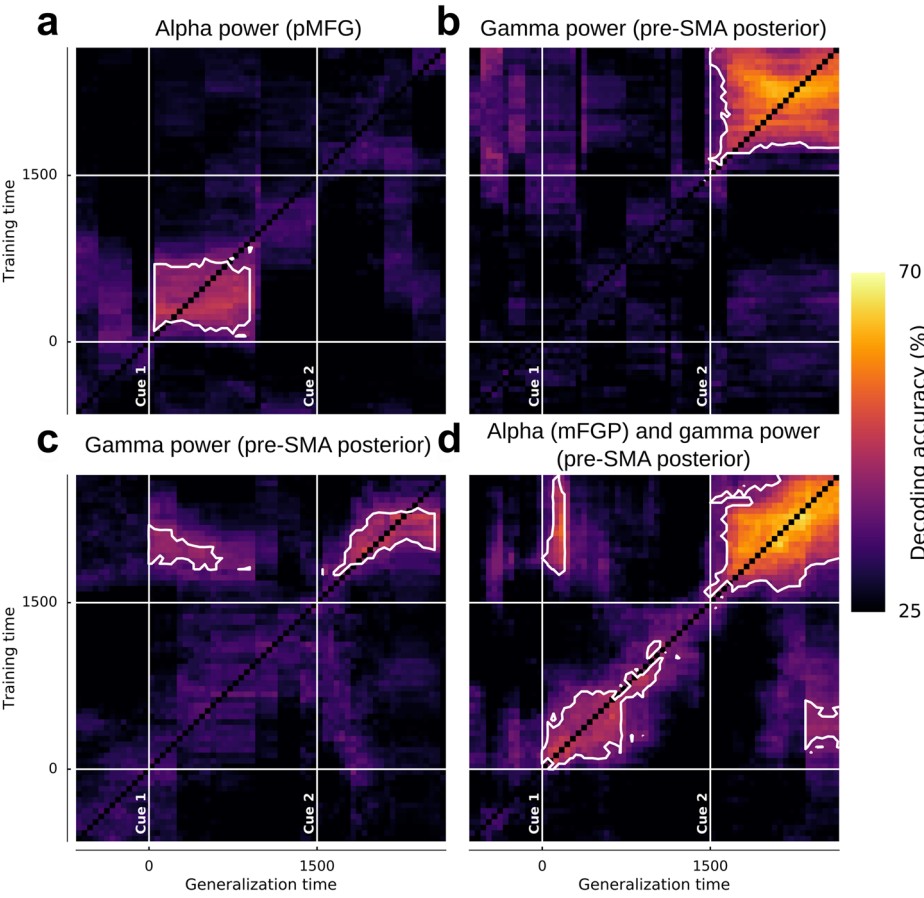

specifically during execution (Fig. 5). Furthermore, we previously reported that PAC varies considerably across motor states, specifically when comparing intention and execution states[28] but whether PAC is differentially modulated across individual movement directions was so far unclear. The results in the present study suggest that PAC varies only weakly across directions and that these modulations are not sufficient for PAC-based single-feature classification of movements. This is consistent with a previous study by Yanagisawa et al. (2012) who showed that PAC differs between motor planning and execution, but not across movement type[25]. Taken together, these data support the idea that PAC in sensorimotor areas may be a large-scale mechanism that constrains gamma activity during rest and planning periods (into slow phasic amplitude fluctuations) and releases it for the purpose of motor execution. Whether other PAC-related metrics (such as the preferred phase, i.e. the phase at which binned amplitude is maximum) might allow for better PAC-based movement decoding still needs to be investigated.

Directional tuning of arm movements in the space of single-unit activity in the motor cortex is widely established textbook knowledge[1]. Moreover, it has also been shown that multi-unit activity and LFP signals can be used to predict movement directions from monkey motor cortex[8–12]. Interestingly, further evidence in macaques has revealed task-related gamma-range LFP modulations in the posterior parietal cortex during both planning and execution of arm and eye movements[13,14]. In humans, movement-type decoding using LFP-based spectral features was also shown in non-primary motor areas[19]. However, the question of whether population activity recorded in brain areas outside the primary motor cortex also exhibits directional tuning has not received much attention. Our findings indicate that movement direction can be inferred from primary and non-primary motor cortices, as well as from non-motor areas including parietal and prefrontal areas. Using single-feature decoding of movement planning, we first found a strong implication of the posterior middle frontal gyrus (pMFG-BA6) especially using alpha

and beta power (Figs. 5, 6, 7a, b) with a maximum decoding of 49.37%. Almost concurrently and using the same frequencies, motor direction planning could also be decoded from the posterior cingulate/precuneus (Figs. 5, 6), which could be consistent with the involvement of this area in internal self-representation[31] and previous reports suggesting that it encodes motor intentions before complete awareness[32,33]. Decoding motor execution was essentially possible through gamma power, essentially in motor-related areas (Figs. 5, 6, 7c, d) and reached a maximum of 62.94%. We also observed that significant decoding via gamma activity in pre-SMA began first in the anterior part followed by the posterior part. Interestingly, the significant movement classification via gamma activity in anterior SMA occurred in the very early stages of the execution, around the go signal, before the decoding in the primary motor cortex (Fig. 6). Finally, the model achieved a modest but above-chance decoding of 44% using the VLFC phase in the posterior pre-SMA during the execution. The VLFC is intricately linked to the readiness potential, a brain signal that arises in motor cortices during voluntary or self-paced movements with a time-locked onset[34,35]. Consequently, it is plausible that the observed decoding accuracy using the VLFC was influenced, at least in part, by varying movement onset times corresponding to each direction.

Movement planning and execution shared a spatially overlapping motor-related network[36]. To address the dynamics of movement direction representation in the brain during planning and execution, we investigated the ability of classifiers to generalize to temporally distant time points using different features. This was achieved using a temporal generalization procedure[37]. Our illustrative analysis (Fig. 8) shows how some premotor sites, involved in externally driven cued movement[38], decode directions only during planning or only during execution (i.e. Fig. 8a, b) while others were able to generalize to both phases (Fig. 8c). Thus, directional tuning of LFPs during planning and execution share partly overlapping neural representations, mainly through alpha and gamma oscillations. However, cross-temporal generalization decoding

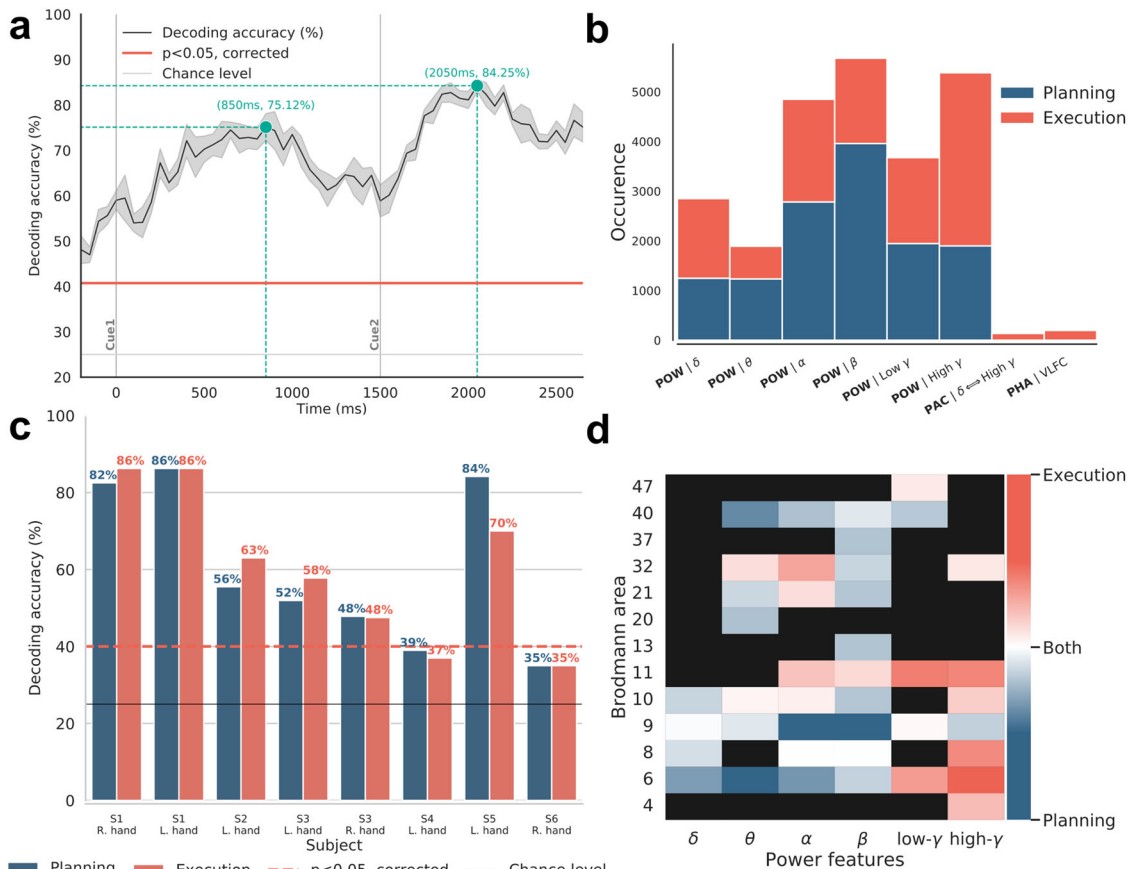

**Fig. 9 | Decoding results of the multi-features procedure. a** Time-resolved decoding accuracy and associated deviation using the MF selection for the subject S1. Cue 1 and Cue 2 are represented with two solid gray lines. Blue lines indicate the maximum decodings reached respectively during the planning and execution periods. The horizontal solid gray line represents the theoretical chance level of 25% and the solid red line is the corrected decoding accuracy ($p < 0.05$ corrected using maximum statistics across time points) obtained by randomly shuffling the label vector (permutations). Shaded areas represent the SEM. **b** Most selected features during planning (blue bars) and execution (red bars). For each barplot, the *y*-axis show the number of times a feature was selected (occurrence), and the *x*-axis shows the feature type (power, phase and PAC) as well as the name of the frequency band.

**c** Best decoding accuracies per subject for intention and execution. The solid black line represents the theoretical chance level of a 4-class classification problem (25%) and the dotted red line (~40%), the statistical chance level at $p < 0.05$ (corrected using maximum statistics across subjects). **d** Most recurrent selected power features during the multi-features procedure as a function of Brodmann area. For each power frequency band and for each Brodmann area, we subtracted the number of features selected during preparation from those during execution. Thus, blue and red colors mean that a feature has been selected more times respectively during the preparation and execution of the movement (specificity). In the same way, the white color means that as many features have been selected for both conditions while black rectangles stand for no selected features.

also showed that directional tuning of LFPs during planning and execution reflects a single and sustained process rather than a dynamically changing coding phenomenon[39]. It has been proposed that pre-SMA acts as an interface to transform visual information into information required for motor planning. Animal studies revealed that pre-SMA neurons encode the spatial location of the target[40]. Therefore, it is possible that the shared information between the execution and planning phase is explained by the target location instead of movement parameters. Furthermore, extending the temporal generalization to multi-feature classifiers (Fig. 8d) illustrates how multi-site feature combinations may lead to models capable of bidirectional generalization (i.e. be trained on execution data and generalize to the planning period, or be trained on the planning and generalize to the execution widow).

We also addressed the decoding complementarity of the spectral features using multivariate classification, where different features from motor and non-motor areas are combined (Fig. 9). Using feature selection within the classification framework allows us to determine the combinations of features that lead to the highest decoding accuracies. By applying this procedure at each time point, we obtained an example of time-resolved decoding reaching 84.25% during the execution and 75.12% during the intention phase (Fig. 9a). Interestingly, in addition to

power features, VLFC phase and delta-gamma coupling were also selected by the algorithm, which suggests complementarity in terms of the directional information these features provide (Fig. 9b). Such spectral features can also be combined to graph-theoretical measurements to further improve decoding performance[41]. Non-surprisingly, the result of multi-features classification varied across subjects highlighting how much the decoding accuracy depends on the intracranial implantation in each patient (Fig. 9c). A general trend was that anterior superior frontal gyrus and temporal implantations did not allow significant decoding, even with multi-feature classification (S4 and S6) while implantations with a majority of middle and medial frontal SEEG sites (S1-2-3-5) allowed significant dissociation up to 86% of the 4-directions for both planning and execution (Fig. 1c). Finally, the data-driven feature selection procedure highlighted the prominent role of the power of slower oscillations (theta, alpha and beta) in the premotor, prefrontal and parietal cortex (BA 6-9-40) for decoding motor intentions. Our findings of directional tuning in BA9 and BA40 is consistent with their role in motor processes, particularly the planning of movement directions[18]. BA40, part of the posterior parietal cortex, is also involved in motor planning processes[14,42–44]. This region is involved in the emulation of sensory information into motor commands, especially during the coding of

**Table 1 | Patient demographics and clinical details: handedness, age, gender, and broad description of epilepsy type as determined by the clinical staff of the Grenoble Neurological Hospital, Grenoble (France)**

|    | Handedness | Age | Gender | Epilepsy | Etiology | EZ localization | Lesion |
|----|-----------|-----|--------|----------|----------|-----------------|--------|
| S1 | R | 19 | F | Frontal | Secondary | Precentral gyrus (RH) | Dysplasia |
| S2 | R | 23 | F | Frontal | Cryptogenic | Precentral gyrus (LH) | Absent |
| S3 | R | 18 | F | Frontal | Cryptogenic | Fronto-basal (RH) | Absent |
| S4 | R | 18 | F | Frontal | Idiopathic | Fronto-central (RH) | Absent |
| S5 | R | 31 | F | Insula | Secondary | Operculum (RH) | Cavernoma |
| S6 | R | 24 | F | Frontal | Secondary | Supra-sylvian posterior (LH) | Vascular sequelae |

Recording sites with epileptogenic activity were excluded from the analyses.

spatial relationships[44,45]. For direction classification during movement execution the feature selection algorithm predominantly used the power in higher frequency bands (low and high gamma) in the motor, premotor, frontal, and cingulate areas (BA 4-6-11-32) (Fig. 9d).

The present study has several limitations that need to be acknowledged. First, intracranial recordings provide high-quality signals at the cost of a heterogeneous and incomplete coverage of the brain. Even with more than 500 recording sites, the probed brain areas were not equally represented in our sample of patients. The implantations across the six subjects (see Fig. 1) yielded a reasonable coverage of frontal (although the right hemisphere was over-represented compared to the left hemisphere) and central areas but the parietal cortex was under-represented. Moreover, four out of six patients had uni-lateral implantations, and the two others had non-symmetrical implantation. Because of this limitation, which is inherent to invasive recordings, it was not possible to separate contralateral from ipsilateral effects on direction decoding. It would be a great benefit to examine whether ipsilateral activity also carries directional information[46]. This could be assessed using group-level statistics on a larger number of patients implanted with intracranial recordings[47] or with EEG or MEG recordings using a similar center-out paradigm. Furthermore, although we systematically excluded electrodes with typical epileptic waveforms (e.g., epileptic spikes), the mere fact that this research was conducted in epilepsy patients may limit the generalizability to healthy subjects. Future studies could investigate the feasibility of reconstructing the continuous 3D hand position using deep learning fitted on depth recordings[48,49]. Finally, we investigated the decoding of only four directions of movement, mainly because of the limited time with the patients. Several studies have addressed the decoding of a higher number of directions during movement execution[30,50,51]. These studies attempted to either decode 8 directions of movements using EEG or ECoG recordings[52,53] or attempted to provide a prediction of a continuous movement using ECoG recordings[11,26,54,55]. However, predicting a continuous trajectory from movement planification signals still remains an open challenge.

## Conclusion

The present study investigated the feasibility of decoding planned and executed limb-movement directions from human intracranial recordings, using a wide range of spectral features (i.e. power, phase, and phase–amplitude coupling in multiple frequency bands and brain areas). We found that decoding during the planning phase mainly involved lower frequencies of power (i.e. alpha and beta) in the posterior middle frontal cortex and parietal areas. We also found significant decoding during movement execution using high-gamma power in motor and premotor areas but also using very low-frequency phase (1.5 Hz). These findings, in addition to the illustrations of the feasibility of multi-feature temporal generalization of directional tuning representation in the human brain, advance our understanding of the role of spectral properties of brain activity in movement planning and control and open up new paths that could be explored in next-generation brain-machine interfaces.

## Methods
### Participants

We collected SEEG recordings from six drug-resistant epilepsy patients (6 females, mean age $22.17 \pm 4.6$, all right-handed). Multi-lead EEG depth electrodes were implanted at the Epilepsy Department of the Grenoble Neurological Hospital (Grenoble, France). Trials containing artefacts or pathological waveforms were systematically excluded through visual inspection in collaboration with the medical staff, as in previous studies[28,56–59]. All participants provided written informed consent, and the experimental procedures were approved by the Institutional Review Board, as well as by the National French Science Ethical Committee. The demographic and clinical details of the patients are summarized in Table 1.

### Electrode implantation and stereotactic EEG recordings

Stereotactic electroencephalography (SEEG) electrodes had a diameter of 0.8, 2 mm wide and 1.5 mm apart (DIXI Medical Instrument®). Each electrode consisted of 10 to 15 contacts according to the implanted structure. This yielded a total of 748 intracerebral sites when pooling the sites of our sample of patients (i.e., 126 sites in each patient, except for one patient who had 118 recording sites). At the time of acquisition, a white matter electrode was used as reference, and data was bandpass filtered from 0.1 to 200 Hz and sampled at 1024 Hz. Electrode locations were determined using the stereotactic implantation scheme and the Talairach and Tournoux proportional atlas[60]. For each subject, electrode location was standardized on Talairach space (based on post-implantation CT). The Talairach coordinates of each electrode were finally converted into the MNI coordinate system according to standard routines[56,61,62] (see Supplementary Table 1). To be able to visualize intracranial recording sites on a 3-D standard MNI brain (Fig. 1a) and to project SEEG data to the nearest cortical surface (Fig. 1b), we used an open-access visualization Python toolbox called Visbrain[63]. Each SEEG site is represented by a color ball into the transparent brain. Cortical projection was obtained by taking the intersection between the cortical surface and a 10 mm radius sphere around each site. Non-significant decoding is systematically turned in gray. By convention, the left hemisphere (LH) is presented on the left in all brain visualizations.

### Experimental design

The experimental paradigm used in this study consisted of a classical delayed center-out motor task[28]. After a rest period of 1000 ms ($-1000$ to 0 ms), the participants were visually cued to prepare a movement towards a visually presented target in one of four possible directions: up, down, left, or right (*Planning phase*, 0–1500 ms). Next, after a 1500 ms delay period, A Go signal, a central cue switching from white to black, prompting the subjects to move the cursor towards the target (*Execution Phase*, 1500–3000 ms) (Fig. 1d).

### Data preprocessing

The data were preprocessed using standard procedures, consistent with our previous intracranial EEG work[39,56,58]. Data recorded in each SEEG site was bipolarized. This procedure consists in subtracting the activity from successive sites in order to remove or reduce artifacts, and increase

the spatial specificity while minimizing the influence of distant sources. Re-referencing each contact to its direct neighbor led to a spatial resolution of ~3 mm[56,64,65]. This bipolarization led to 580 bipolar derivations across all subjects. Finally, trials contaminated by epileptic activity and electrodes located close to the extra-ocular eye muscles were removed from the analyses by visual inspection of the time-series and time–frequency decomposition and insights from the clinical staff. The final number of trials retained for analyses across patients varied between 120 and 360 (215 ± 77).

### Spectral analyses

A wide range of oscillatory brain features (power, phase and phase-amplitude coupling) were extracted from the SEEG recording using the Hilbert transform. To this end, we first filtered the data in the required band using a two-way zero-phase lag finite impulse response (FIR) Least-Squares filter implemented in the EEGLAB toolbox[66]. Then, phase and amplitude components were computed using the Hilbert transform on filtered data. The following frequency bands were considered: very low-frequency component (VLFC) [0.1; 1.5 Hz], delta ($\delta$) [2–4 Hz], theta ($\theta$) [5–7 Hz], alpha ($\alpha$) [8–13 Hz], beta (–) [13–30 Hz], low-gamma (low $\gamma$) [30–60 Hz] and broadband gamma (high $\gamma$) [60-200 Hz]. Power features were computed for $\delta, \theta, \alpha, \beta$, low-$\gamma$ and high-$\gamma$, while phase values were extracted for VLFC, $\delta, \theta$, and $\alpha$. PAC was computed between $\delta, \theta$, and $\alpha$ phases and high-$\gamma$ amplitude. Furthermore, in order to investigate the time course of decoding performance, we systematically considered 67 points across time. The choice of temporal resolution/windows was different and will be described in their respective sub-sections. Eventually, each feature involved 67 time points. For each SEEG site, 13 features were calculated (6 of power, 4 of phase and 3 of PAC) with 67 time points. Across all SEEG sites, this led to a total of 505180 ((6 + 4 + 3) * 67 * 580) independent features to classify.

**Instantaneous power features estimation.** From the band-specific Hilbert transform, power modulations were computed by taking the square of time resolved amplitude. For the specific case of the high-gamma band, the [60, 200 Hz] was split into 10 Hz non-overlapping sub-bands, and final gamma power modulations were obtained by taking the mean of those multiple sub-bands, according to previous routines[56,58,62,67–70]. Power was averaged using a 700 ms sliding window, with a 50 ms shifting, leading to 67 time points. The classification was applied to unnormalized power. We applied a normalization only for the specific case of the visualization (time-frequency maps and single trial representation, see Fig. 2). To this end, to each frequency band, we subtracted then divided by the mean of a 500 ms baseline window, centered according to the pre-stimulus rest period ([−750, −250 ms]).

**Instantaneous phase features estimation.** For a specific frequency band, phase features were extracted from the angle of the Hilbert transform. For classification, we selected a point every 50 ms from this instantaneous phase. Finally, we used Rayleigh's test to estimate significant phase modulations[71–73], using the circular statistics toolbox[74]. This instantaneous phase is then used for the classification. To observe phase-alignment consistency across trials, we compute the Phase Locking Factor, defined as the mean across the modulus of a single trial phase[71] (Fig. 3). In order to have consistency with power features, we selected the instantaneous phase at each center of above defined 700 ms power time windows, which led to 67 phase points across time.

**Phase–amplitude coupling features estimation.** First, the filter order for extracting phase and amplitude was systematically adapted, using three cycles of slow oscillations (for phase) and six cycles for amplitude[75]. PAC estimations can be estimated by a large variety of measures[76–80]. We tested several of them, mainly the mean vector length (MVL)[81] and the Kullback–Leiber divergence (KL)[78]. Both methods yielded similar results, but after slightly adapting the MVL, we obtained PAC estimation leading to better decoding accuracies compared to the KL method. In order to

improve PAC robustness, we generated surrogates by randomly swapping phase and amplitude trials[78]. Then, the original modulus is z-scored normalized using the mean and the deviation of 200 generated surrogates (Fig. 4). PAC algorithms used in this paper are all implemented into an open-source Python package called Tensorpac[82]. The PAC was estimated using the same windows as power features, meaning windows of length of 700 ms shifted every 50 ms, which led to the same number of 67 windows.

### Signal classification

We explored the feasibility of time-resolved direction decoding from human LFP using two strategies of increasing dimensionality: (a) a single feature approach to evaluate the performance of each feature, (b) an inter-site and inter-feature approach using a feature selection procedure to estimate the final decoding using all available intracranial EEG recordings. These two strategies are performed at each of the 67 time points defined above, providing an overview of which feature, where and when they are decoding, and how reliable they are. In contrast to brain decoding approaches with non-invasive brain recordings (e.g. EEG or MEG), inter-subject cross-validation is not straightforward for SEEG since electrode implantations differ across subjects. We thus performed intra-subject cross-validation[28]. All classifications were implemented in Python 3 using the sci-kit-learn package[83].

**Single feature evaluation.** We classified each feature at each of the 67-time windows defined above for all subjects and all recording sites. We compared the performance of several classification algorithms (linear discriminant analysis (LDA), Naïve Bayes (NB), $k$th nearest neighbor (KNN), and support vector machine (SVM) with linear and radial basis function (RBF) kernels. They all provided similar performances, and we finally chose the LDA for its efficiency. The performance of the classifier was evaluated by computing the % decoding accuracy (i.e. proportion of successfully classified samples in the test set), which was obtained following a standard stratified 10 times 10-fold cross-validation scheme. To assess the statistical significance of the decoding performances, we used permutation testing to generate null distributions by randomly shuffling the data labels[84,85]. Correction for multiple comparisons was assessed by generating a distribution of permutation maxima across time, space, and frequency (i.e. maximum statistics)[86,87].

**Cross-temporal generalization of classification.** To evaluate whether a classifier trained during execution can be used to decode movement directions during preparation (or vice versa), we used a temporal generalization (TG) procedure[37]. In principle, a classifier is trained at a particular moment in the task (training time axe) and then tested at another time (testing time axe). Note that we performed TG using both single and multi-feature classification.

**Multi-feature classifications using feature selection.** To identify groups of features that jointly lead to higher decoding performances compared to single-feature classification results, we used multi-feature (MF) classification. For each time sample, the MF procedure determines the best possible combination across all feature types (power, phase, and phase–amplitude coupling) and across all SEEG recording sites per subject. We combined a wrapper method[88–90] (Select $k$-best, with $k$ between [1,10]) with a filter method[91,92] (false discovery rate, FDR with a type 1 error rate of 5%) which are respectively the SelectKBest and the SelectFdr functions of scikit-learn[83]. The MF classification was performed using a linear SVM for the whole MF pipeline. As recommended, we linearly rescaled each attribute to be zero mean with a unit variance[93]. Multi-feature pipeline: To estimate MF performances, we implemented the following pipeline: (1) A first 10-fold cross-validation was defined to generate a training and testing set, (2) the training dataset is used to fit parameters of the transformation for data rescaling, then, this set is rescaled. (3) We optimized the number of selected features for the k-best using a 3-fold cross-validated grid search. We then took the union of selected features determined by the $k$-best and FDR and got a reduced

version of our training set, (4) we trained the linear SVM on this optimal training set, (5) the testing set is rescaled with the same parameters used for the training set. Then, the selected attributes of the training set are used to select those on the testing set, (6) the already trained classifier was finally tested to predict labels on this optimal testing set and turn this prediction into decoding accuracy. For the statistical evaluation, this whole pipeline is embedded in a loop of 200 occurrences where, for each occurrence, the label vector is shuffled. Those 200 permutations allow statistical assessments with $p$-values as low as 0.005.

## Statistics and reproducibility
All data were analyzed using custom Python code. Statistical analysis was performed using non-parametric tests.

## Reporting summary
Further information on research design is available in the Nature Portfolio Reporting Summary linked to this article.

## Data availability
Raw data cannot be shared due to ethics committee restrictions related to the clinical acquisition setting. Intermediate as well as final processed data that support the findings of this study are available from the corresponding author (E.C.) upon reasonable request.

## Code availability
The custom codes used to generate the figures and statistics are available from the lead contact (E.C.) upon request.

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

## Acknowledgements

E.C. was supported in part by a Ph.D. Scholarship from the Ecole Doctorale Inter-Disciplinaire Sciences-Santé (EDISS), Lyon, France, and funding from the Natural Sciences and Engineering Research Council of Canada (NSERC). We acknowledge support from the Brazilian Ministry of Education (CAPES grant 1719-04-1), the Foundation pour la Recherche Médicale (FRM, France), and the Fulbright Commission to J.L.P.S. K.J. was supported by funding from the Canada Research Chairs program and a Discovery Grant (RGPIN-2015-04854) from the Natural Sciences and Engineering Research Council of Canada (NSERC), a New Investigators Award from the Fonds de Recherche du Québec—Nature et Technologies (2018-NC-206005) and an IVADO-Apogée fundamental research project grant. The authors are grateful for the collaboration of the patients and clinical staff at the Epilepsy Department of the Grenoble University Hospital. The operation of the supercomputer is funded by the Canada Foundation for Innovation (CFI), the Ministère de l'Économie, de la Science et de l'Innovation du Québec (MESI) and the Fonds de Recherche du Québec—Nature et technologies (FRQNT). Computations were made on the supercomputer Guillimin from the University of Montréal, managed by Calcul Québec and Compute Canada. The operation of this supercomputer is funded by the Canada Foundation for Innovation (CFI), the ministère de l'Économie, de la science et de l'innovation du Québec (MESI) and the Fonds de recherche du Québec—Nature et technologies (FRQ-NT). Computations were made on the supercomputer Guillimin from the University of Montréal, managed by Calcul Québec and Compute Canada.

## Author contributions

E.C.: Conceptualization, software, formal analysis, visualization, writing—original draft; F.D.R.: Writing—original draft and review & editing; A.-L.S.: Conceptualization, Software, Methodology; M.P.-B.: Resources; J.LP.S.: Methodology; P.K.: Resources; J-P.L.: Resources; A.G.: Conceptualization, supervision, writing—original draft and review & editing; K.J.: Conceptualization, writing—original draft and review & editing, supervision, project administration, funding acquisition.

## Competing interests

The authors declare no competing interests.
