## [Peer Review File · Communications Biology]

Reviewers' comments:

Reviewer #1 (Remarks to the Author):

The authors examined whether intra-cortical LFPs from six epilepsy patients were predictive of movement directions. They used power, phase and phase-amplitude coupling in the usual bands, from delta to broadband gamma (high gamma) [60-200Hz].

Power features computed for delta, theta, alpha, beta, low-g, high-g, while phases were extracted for VLPFC, delta, theta and alpha. PAC was computed between delta, theta, alpha and high-gamma.

Main comment

I indeed found the manuscript very interesting. I agree with the authors about the relevance of LFPs to decode movements. They are most certainly complementary to SUA.

Under the conditions performed by their experimental task, the result is clear. Furthermore, it is the first case in which the relative contribution of amplitude, phase and PAC features for decoding planned and limb movement direction is tested with humans. It is also of relevant to provide significant insight onto the decoding of directional tuning in areas other than premotor and primary motor areas, thus providing significant support to a distributed representation across pMFG in particular SMA/pre-SMA.

Major Issue

My first concern is about the data itself. I sympathise with the difficulty of gathering data from epilepsy patients, but you need to provide precise electrode locations per participant (beyond what you have shown), which I clearly miss in this study, as well as a detailed account of the generalization procedure used to combine data across participants and its limitations. This casts some doubt as to the generalization of these results.

My second concern is the cross-Temporal Generalization procedure used for validation. The use of this procedure for validation is not properly justified, as it is based on the notion that the elements in the data the encoder works upon extend throughout the entire interval of the planning/movement phase, which is a big if. It is not the most common procedure, please, provide a proper explanation. The choice I guess is justified by the nature of the information being decoded, and it is clear that it would not work to decode movement kinematics of any time dependent movement. I would suggest a partial comparison with a more classic 80-20% training-testing dataset approach as an additional validation in the most fundamental cases.

I think the manuscript provides an interesting insight on human motor coding and should therefore be published. However, it needs to be rewritten in such a way that it dissipates any doubt as to why/how is data from multiple participants can be combined and analysed in this unusual fashion, and about the fundamental technique you have used to derivate your results.

Reviewer #2 (Remarks to the Author):

The manuscript describes the possibility of movement direction decoding during planning (alone or in addition to execution) with oscillation phase and cross-frequency interactions. Because of the richness of data in number of areas involved and the 3-fold analysis of single-feature classification, temporal dynamics and multivariate classification it is an important paper for insight on movement dynamics and improvement of upper limb BCIs.

It is well structured and results are at a higher level of detail than most studies in this area. They managed to convey both positive (direction decoding is possible in the planning phase) and negative results (phase and PAC are in most cases uninformative) in a well written manuscript.

I have some remaining comments:

- 1) P4: table 1 uses subject names P1, etc. figure 1 and text uses S1. Please use the same code in all tables, figures and text.
- 2) P6: For classification, we selected of point every 50ms from this instantaneous phase. -> Please correct sentence.
- 3) P6: after slightly adapted the MVL -> elaborate on the adaptation (adapted-> adapting)
- 4) p7: (b) an inter-site and inter-feature using a feature selection procedure -> (b) an inter-site and inter-feature approach using a feature selection procedure
- 5) P9 High-gamma (60-200 Hz) power led to 62.94% ($p < 0.05$) correct classification during the execution in the posterior pre-SMA decoding results clearly surpassed those obtained in the lower-frequency bands. -> please rephrase this sentence for readability.
- 6) P12: Intriguingly, VLFC phases showed the highest differences during the execution at approximately the same time as gamma power - - the VLFC power signal may be caused by the same broadband effect as the high gamma band. If so, it is surprising that phase could, the authors should elaborate on this, e.g. in the discussion.
- 7) P13: if some SEEG sites were able to decode during the execution while training the classifier during the planning period and conversely -> to me this research question does not fit in the overall structure of the manuscript. I suggest to add to the introduction why the authors feel this is an important question for upper limb decoding in BCI. The rationale behind this question in the discussion (multi-site feature combinations may lead to models capable of bidirectional generalization) is not satisfactory.
- 8) P18 The authors chose to classify four directions: up down left right. A limitation of the study is this relatively low number of directions. I assuming that this limit was imposed due to restricted patient time, but I would like to see some discussion on this limit and suggestions on expected results with more directions.

Reviewer #3 (Remarks to the Author):

In the article “Human local field potentials in motor and non-motor brain areas encode upcoming movement direction”, the authors aimed to decode movement planning during motor movement. For this, in 6 epilepsy patients local field potentials were recorded using SEEG. The power, phase, and phase-amplitude coupling were measured at different frequency bands. Power from six frequency bands whereas the phase at four and phase-amplitude coupling only at three frequency bands. In total 13 features were extracted and applied to two different machine learning algorithms namely LDA for single feature and SVM for multifeatured classification. The current study offers new insights using sEEG recordings for decoding movement execution and planning in epilepsy patients. There are methodological concerns, which need clarification as pointed out below:

Comments to the authors

1. The selection of frequency bands for each of the measures that need to be motivated was surprising that the beta frequency was not used.
2. How many trials per patient should be mentioned after artifact removal? In the figures, 40 trials are shown.
3. In all the figures mentioned the plots are from all the patients (grand average) or selected patients.
4. The signal classification is confusing the authors mention that they tested several machine learning algorithms and finally chose LDA. So, for a single feature, the LDA, and for the multi-feature selection SVM was used? If yes justification needs to be provided.
5. In the single feature section explain clearly how the multiple corrections were done taking the 99 percentile of the distribution or based on what?
6. The reasoning for having good decoding accuracy for planning and execution for patient S1 is clear but not S2 is puzzling as shown in Figure 9. At the same time, S5 based on Figure 1 also shows good accuracy due to frontal electrodes.
7. The VLF decoding could be related to the “readiness potential” phenomenon worth discussing about it.

Reviewer #1

The authors examined whether intra-cortical LFPs from six epilepsy patients were predictive of movement directions. They used power, phase and phase-amplitude coupling in the usual bands, from delta to broadband gamma (high gamma) [60-200Hz].

Power features computed for delta, theta, alpha, beta, low-g, high-g, while phases were extracted for VLPFC, delta, theta and alpha. PAC was computed between delta, theta, alpha and high-gamma.

Main comment

I indeed found the manuscript very interesting. I agree with the authors about the relevance of LFPs to decode movements. They are most certainly complementary to SUA.

Under the conditions performed by their experimental task, the result is clear. Furthermore, it is the first case in which the relative contribution of amplitude, phase and PAC features for decoding planned and limb movement direction is tested with humans. It is also of relevant to provide significant insight onto the decoding of directional tuning in areas other than premotor and primary motor areas, thus providing significant support to a distributed representation across pMFG in particular SMA/pre-SMA.

We thank this reviewer for the positive feedback about the manuscript. Please find below our detailed answers to all the issues raised.

Major Issue

My first concern is about the data itself. I sympathise with the difficulty of gathering data from epilepsy patients, but you need to provide precise electrode locations per participant (beyond what you have shown), which I clearly miss in this study, as well as a detailed account of the generalization procedure used to combine data across participants and its limitations. This casts some doubt as to the generalization of these results.

We agree with the reviewer that it is important in the case of iEEG data to provide a detailed description of the recording sites. This was done in the previous version of the paper in panels A, B and C of figure 1, which provides three ways of appreciating the spatial coverage across all six participants. But we understand the interest of providing an even more detailed description of the electrode locations. We have now added a table that lists all the individual electrode coordinates across all 6 subjects in MNI space (Supplementary Table 1).

Regarding population-level inferences, we agree with the reviewer that this is probably the biggest challenge when working with intracranial EEG recordings because of the sparse spatial sampling. Here, as described in the method section, we are performing decoding and significance testing at the single-recording and single-subject levels. We then combine the results from all the subjects by projecting on the cortical surface the significant decoding obtained at the single-subject level. A proper group-level analysis as it is performed using with spatially uniform recordings like EEG and MEG was not possible here, also because of the relatively small number of subjects. We added the following sentence to the “*Limitations and future paths*” of the discussion section, with a reference :

This could be assessed using group-level statistics on a larger number of patients implanted with intracranial recordings (Combrisson et al., 2022), or with EEG or MEG recordings using a similar center-out paradigm.

My second concern is the cross-Temporal Generalization procedure used for validation. The use of this procedure for validation is not properly justified, as it is based on the notion that the elements in the data the encoder works upon extend throughout the entire interval of the planning/movement phase, which is a big if. It is not the most common procedure, please, provide a proper explanation. The choice I guess is justified by the nature of the information being decoded, and it is clear that it would not work to decode movement kinematics of any time dependent movement . I would suggest a partial comparison with a more classic 80-20% training-testing dataset approach as an additional validation in the most fundamental cases.

We thank the reviewer for raising this interesting question. To be clear, the manuscript is subdivided into three questions. First, we tried to decode movement directions during both the planning and execution phases. To tackle this question, we used as the reviewer suggested a classical 10-fold (90-10% training-testing split), repeated 10 times for robustness. Then we wondered whether the representation of movement directions during planning was similar to the representation of movement direction during the execution. To tackle this question, we used the cross-temporal generalization which consists of training a decoder to recognize directions during planning (training set) but then predict directions using the data coming from the execution phase (testing set), and conversely. Therefore, training and testing sets are distinct by construction. As pointed by the reviewer, this generalization is only possible if the representation of the directions is shared between the two phases. But this is the hypothesis we were testing. The third question addresses the decoding capabilities of intracranial recordings by combining feature types at different spatial locations.

The lack of clarity was also raised by the reviewer #2. As a consequence, we added the following sentence to the introduction of the paper:

To address these gaps we investigated whether the classification of arm movement directions was possible using phase, amplitude and Phase-Amplitude Coupling (PAC) features during both planning and execution. We then asked whether movement directions share common neural representations during both phases. To tackle this question, we trained a classifier at execution and tested whether it was able to decode movement directions at planning and conversely. Investigating the cross-temporal generalizations in both directions was aimed at probing similarities of neural representations of movement directions between planning and execution.

We also improved the first sentence in the result section "*Temporal Generalization of the decoding of movement directions*":

We then tested whether movement direction representation was shared between planning and execution. To this end, we used TG using either single or multiple power features (e.g., alpha alone or in combination with gamma) to find if some SEEG sites were able to decode during the execution while training the classifier during the planning period and conversely

I think the manuscript provides an interesting insight on human motor coding and should therefore be published. However, it needs to be rewritten in such a way that it dissipates any doubt as to why/how is data from multiple participants can be combined and analysed in this unusual fashion, and about the fundamental technique you have used to derivate your results.

We agree with the reviewer, and we hope that the revised version reasonably achieves most of these requirements

Reviewer #2

The manuscript describes the possibility of movement direction decoding during planning (alone or in addition to execution) with oscillation phase and cross-frequency interactions. Because of the richness of data in number of areas involved and the 3-fold analysis of single-feature classification, temporal dynamics and multivariate classification it is an important paper for insight on movement dynamics and improvement of upper limb BCIs.

It is well structured and results are at a higher level of detail than most studies in this area. They managed to convey both positive (direction decoding is possible in the planning phase) and negative results (phase and PAC are in most cases uninformative) in a well written manuscript.

We thank the reviewer for this positive feedback. Please find below our answers regarding the main points to address.

I have some remaining comments:

1) P4: table 1 uses subject names P1, etc. figure 1 and text uses S1. Please use the same code in all tables, figures and text.

We corrected the subject names in table 1.

2) P6: For classification, we selected of point every 50ms from this instantaneous phase. -> Please correct sentence.

We corrected the typo in the revised manuscript.

3) P6: after slightly adapted the MVL -> elaborate on the adaptation (adapted-> adapting)

We corrected the typo. We also added the following sentence to clarify our adaptations:

In the original version of the MVL, the surrogates were computing using a bloc-swapping approach. Instead, here the surrogates were generated by randomly swapping phase and amplitude trials ⁵².

4) p7: (b) an inter-site and inter-feature using a feature selection procedure -> (b) an inter-site and inter-feature approach using a feature selection procedure

We added the missing word in the revised manuscript.

5) P9 High-gamma (60-200 Hz) power led to 62.94% ($p < 0.05$) correct classification during the execution in the posterior pre-SMA decoding results clearly surpassed those obtained in the lower-frequency bands. -> please rephrase this sentence for readability.

We thank the reviewer for pointing us toward the missing punctuation. We corrected it in the revised manuscript. The sentence now reads:

High-gamma (60-200 Hz) power led to 62.94% ($p < 0.05$) correct classification during the execution in the posterior pre-SMA. Decoding Results clearly surpassed those obtained in the lower-frequency bands.

6) P12: Intriguingly, VLFC phases showed the highest differences during the execution at approximately the same time as gamma power - - the VLFC power signal may be caused by the same broadband effect as the high gamma band. If so, it is surprising that phase could, the authors should elaborate on this, e.g. in the discussion.

We thank the reviewer for this question, however here, we are talking about the phase of the very low frequency component (VLFC), not its amplitude. To the best of our knowledge, the only mechanism that could potentially link the phase to the amplitude is the phase-amplitude coupling (PAC) that is used here. However, in our study it was not possible to investigate the PAC between the VLFC phase and the gamma power because the minimal single trial window size which would be required for this surpasses the length of our single trial windows. More specifically: the PAC is a measure estimated across time points, because it quantifies the degree of covariation between low-frequency phase and high-frequency amplitude. Therefore, the longer the time-series, the better the PAC estimations because longer time-series provide more cycles to quantify the coupling. A rule of thumb is to use at least four cycles of phase and eight cycles of amplitude. The VLFC phase is extracted between [0.1Hz; 1.5Hz] because raw data are bandpass filtered at the acquisition time. Four cycles of VLFC means that the time window would last at least 2.5 seconds which is much longer than the length of our trials. Finally, it is questionable that such coupling could have a functional motor relevance considering that short and fast movements are often required, as it is the case here. This said, we thank the reviewer for this observation and opportunity to clarify this point.

7) P13: if some SEEG sites were able to decode during the execution while training the classifier during the planning period and conversely -> to me this research question does not fit in the overall structure of the manuscript. I suggest to add to the introduction why the authors feel this is an important question for upper limb decoding in BCI. The rationale behind this question in the discussion (multi-site feature combinations may lead to models capable of bidirectional generalization) is not satisfactory.

We agree with the reviewer that the research question behind the cross-temporal generalization was not clear in the original manuscript. It is true that from a BCI perspective, it is more crucial to examine whether we can infer movement intentions from the planning period. From a basic neuroscience approach, exploring both directions (which are offered naturally by the temporal-generalization methodology) allows for a more general investigation of the similarities of neural representations between execution and

planning. We believe some readers will find these observations of interest, but we do not expand too much on this. To take this reviewer's comment into account, we added the following clarification sentence to the introduction:

To address these gaps we investigated whether the classification of arm movement directions was possible using phase, amplitude and Phase-Amplitude Coupling (PAC) features during both planning and execution. We then asked whether movement directions share common neural representations during both phases. To tackle this question, we trained a classifier at execution and tested whether it was able to decode movement directions at planning and conversely. Investigating the cross-temporal generalizations in both directions was aimed at probing similarities of neural representations of movement directions between planning and execution.

We also improved the first sentence in the result section "*Temporal Generalization of the decoding of movement directions*":

We then tested whether movement direction representation was shared between planning and execution. To this end, we used TG using either single or multiple power features (e.g., alpha alone or in combination with gamma) to find if some SEEG sites were able to decode during the execution while training the classifier during the planning period and conversely

8) P18 The authors chose to classify four directions: up down left right. A limitation of the study is this relatively low number of directions. I assuming that this limit was imposed due to restricted patient time, but I would like to see some discussion on this limit and suggestions on expected results with more directions.

We thank the reviewer for this interesting suggestion. The "*Limitations and future paths*" paragraph of the discussion section already contained the following sentence: "*Finally, future studies could investigate the feasibility of reconstructing the continuous 3D hand position using deep learning fitted on depth recordings*^{83,84}." In the revised manuscript, we extended this discussion about decoding more directions :

Finally, we investigated the decoding of only four directions of movement, mainly because of the limited time with the patients. Several studies have addressed the decoding of a higher number of directions during movement execution (Gunduz and Schalk, 2018; Jerbi et al., 2011; Tam et al., 2019). These studies attempted to either decode 8 directions of movements using EEG or ECoG recordings (Wang et al., 2012; Wolpaw and McFarland, 2004) or attempted to provide a prediction of a continuous movement using ECoG recordings (Hammer et al., 2013; Pistohl et al., 2008; Schalk et al., 2008, 2007). However, predicting a continuous trajectory from movement planification signals still remains an open challenge.

We thank this reviewer for these comments which have improved the quality of our manuscript.

Reviewer #3

In the article "Human local field potentials in motor and non-motor brain areas encode upcoming movement direction", the authors aimed to decode movement planning during motor movement. For this, in 6 epilepsy patients local field potentials were recorded using SEEG. The power, phase, and phase-amplitude coupling were measured at different frequency bands. Power from six frequency bands whereas the phase at four and phase-amplitude coupling only at three frequency bands. In total 13 features were extracted and applied to two different machine learning algorithms namely LDA for single feature and SVM for multifeatured classification. The current study offers new insights using sEEG recordings for decoding movement execution and planning in epilepsy patients. There are methodological concerns, which need clarification as pointed out below:

Comments to the authors

1. The selection of frequency bands for each of the measures that need to be motivated was surprising that the beta frequency was not used.

We thank the reviewer for this interesting comment. Indeed, here we used the beta power only and it showed significant encoding of movement directions, both during the planning and execution phases. However, we did not include beta/high-gamma phase-amplitude coupling because of the proximity of the potential harmonic occurring between the upper bound of the beta band (30Hz) and the lower bound of the gamma band (60Hz). For the same reason, we did not include the low-gamma/high-gamma PAC. As we excluded those two PAC features, we also removed them from the phase features. However, to be clear, we did report that beta power shows significant decoding, as would be expected from previous research.

2. How many trials per patient should be mentioned after artifact removal? In the figures, 40 trials are shown.

We thank the reviewer for pointing us toward this missing information. We added the following sentence to the revised manuscript:

Finally, trials contaminated by epileptic activity and electrodes located close to the extra-ocular eye muscles were removed from the analyses by visual inspection of the time-series and time-frequency decomposition and insights from the clinical staff. The final number of trials retained for analyses across patients varied between 120 and 360 (215±77).

3. In all the figures mentioned the plots are from all the patients (grand average) or selected patients.

As stated in the text, figures 2, 3, 4, 7 and 8 are illustrative figures coming from individual participants. The cortical areas covered across all participants is represented in Fig 1B, while Fig 5 and 9 are population-level results.

4. The signal classification is confusing the authors mention that they tested several machine learning algorithms and finally chose LDA. So, for a single feature, the LDA, and for the multi-feature selection SVM was used? If yes justification needs to be provided.

The reviewer is right. We indeed used the LDA for single-feature classification and SVM for multi-features classification. This choice was encouraged by computational requirements. Considering all types of features, bipolar derivations, and sliding windows, we had a total of 505 180 independent features to classify. For robustness, we performed a 10-times 10-folds cross-validation and 1000 permutations for each feature for significance testing. In total, this represents $\sim 5 \times 10^{10}$ individual classifications to perform. Given this number, we choose the LDA for its efficiency. The multi-features pipeline relies on prior steps to select the most relevant features. Therefore, it required fewer individual classifications which allowed us to use a slightly more elaborate model.

5. In the single feature section explain clearly how the multiple corrections were done taking the 99 percentile of the distribution or based on what?

We thank the reviewer for this remark, however, here we used a very classical approach for significance testing using a non-parametric permutation-based approach. We are not taking the 99 percentile. Here, the correction for multiple comparisons (MCP) is performed using the maximum statistics approach. In short, we try to predict hand direction using each feature and the null distribution is obtained by shuffling the label vector 1000 times and recomputing the classification. To correct across time, space and frequency, we generated a distribution of 1000 elements composed of the maxima of all of the permutations. This distribution of maxima is finally used to infer the p-value corrected for MCP, as explained in the main text: *"To assess the statistical significance of the decoding performances we used permutation testing to generate null distributions by randomly shuffling the data labels^{58,59}. Correction for multiple comparisons was assessed by generating a distribution of permutation maxima across time, space and frequency (i.e. maximum statistics)^{60,61}."*

6. The reasoning for having good decoding accuracy for planning and execution for patient S1 is clear but not S2 is puzzling as shown in Figure 9. At the same time, S5 based on Figure 1 also shows good accuracy due to frontal electrodes.

We thank the reviewer for this relevant comment. Human intracranial EEG offers the best possible signal-to-noise ratio in humans, allowing within-subject inferences. However, a limiting factor is the fact that the implantation of the electrodes is driven uniquely by clinical considerations in an attempt to maximize the chances of localizing the seizure onset zone. Taken together, it is impossible to achieve similar decoding performance between individuals as it depends on the localization of the electrodes, contrary to scalp

EEG or MEG recordings. This topic is discussed in the "*Limitations and future path*" paragraph of the discussion section.

7. The VLF decoding could be related to the "readiness potential" phenomenon worth discussing about it.

We are thankful for the reviewer's suggestion. This is an interesting and noteworthy point. We have now added the following paragraph to the discussion section:

Finally, the model achieved a modest but above-chance decoding of 44% using the VLFC phase in the posterior pre-SMA during the execution. The VLFC is intricately linked to the readiness potential, a brain signal that arises in motor cortices during voluntary or self-paced movements with a time-locked onset (Schurger et al., 2021; Shibasaki and Hallett, 2006). Consequently, it is plausible that the observed decoding accuracy using the VLFC was influenced, at least in part, by varying movement onset times corresponding to each direction.

We thank this reviewer for the helpful suggestions and insightful comments.

References

- Gunduz, A., Schalk, G., 2018. Ecog-based bcis. *Brain-Computer Interfaces Handb.* 297–322.
- Hammer, J., Fischer, J., Ruescher, J., Schulze-Bonhage, A., Aertsen, A., Ball, T., 2013. The role of ECoG magnitude and phase in decoding position, velocity, and acceleration during continuous motor behavior. *Front. Neurosci.* 7. <https://doi.org/10.3389/fnins.2013.00200>
- Jerbi, K., Vidal, J.R., Mattout, J., Maby, E., Lecaigard, F., Ossandon, T., Hamamé, C.M., Dalal, S.S., Bouet, R., Lachaux, J.-P., Leahy, R.M., Baillet, S., Garnero, L., Delpuech, C., Bertrand, O., 2011. Inferring hand movement kinematics from MEG, EEG and intracranial EEG: From brain-machine interfaces to motor rehabilitation. *IRBM* 32, 8–18. <https://doi.org/10.1016/j.irbm.2010.12.004>
- Pistohl, T., Ball, T., Schulze-Bonhage, A., Aertsen, A., Mehring, C., 2008. Prediction of arm movement trajectories from ECoG-recordings in humans. *J. Neurosci. Methods* 167, 105–114. <https://doi.org/10.1016/j.jneumeth.2007.10.001>
- Schalk, G., Kubanek, J., Miller, K., Anderson, N., Leuthardt, E., Ojemann, J., Limbrick, D., Moran, D., Gerhardt, L., Wolpaw, J., 2007. Decoding two-dimensional movement trajectories using electrocorticographic signals in humans. *J. Neural Eng.* 4, 264.
- Schalk, G., Miller, K.J., Anderson, N.R., Wilson, J.A., Smyth, M.D., Ojemann, J.G., Moran, D.W., Wolpaw, J.R., Leuthardt, E.C., 2008. Two-dimensional movement control using electrocorticographic signals in humans. *J. Neural Eng.* 5, 75–84. <https://doi.org/10.1088/1741-2560/5/1/008>
- Schurger, A., Pak, J., Roskies, A.L., others, 2021. What is the readiness potential? *Trends Cogn. Sci.* 25, 558–570.
- Shibasaki, H., Hallett, M., 2006. What is the Bereitschaftspotential? *Clin. Neurophysiol.* 117, 2341–2356. <https://doi.org/10.1016/j.clinph.2006.04.025>
- Tam, W., Wu, T., Zhao, Q., Keefer, E., Yang, Z., 2019. Human motor decoding from neural signals: a review. *BMC Biomed. Eng.* 1, 22. <https://doi.org/10.1186/s42490-019-0022-z>
- Wang, Z., Gunduz, A., Brunner, P., Ritaccio, A.L., Ji, Q., Schalk, G., 2012. Decoding onset and direction of movements using Electrocorticographic (ECoG) signals in humans. *Front. Neuroengineering* 5. <https://doi.org/10.3389/fneng.2012.00015>
- Wolpaw, J.R., McFarland, D.J., 2004. Control of a two-dimensional movement signal by a noninvasive brain-computer interface in humans. *Proc. Natl. Acad. Sci. U. S. A.* 101, 17849–17854.

REVIEWERS' COMMENTS:

Reviewer #1 (Remarks to the Author):

All major elements raised in the previous review have been properly answered and addressed by the authors. The new manuscript shows significant improvement and yields a clearer message. I recommend the manuscript to be published.

Reviewer #2 (Remarks to the Author):

I have no further comments.

Reviewer #3 (Remarks to the Author):

The authors have addressed all the comments raised by me in the revised version of the manuscript do not have further comments.